# Advanced Research on Polymer Floating Carrier Application in Activated Sludge Reactors

**DOI:** 10.3390/polym14132604

**Published:** 2022-06-27

**Authors:** Nikolay Makisha

**Affiliations:** Research and Education Centre “Water Supply and Wastewater Treatment”, Moscow State University of Civil Engineering, 26, Yaroslaskoye Highway, 129337 Moscow, Russia; makishana@mgsu.ru

**Keywords:** wastewater treatment, biofilm carrier, nutrient removal, BOD, floating carrier, filling ratio

## Abstract

This research estimates the efficiency of domestic wastewater treatment in the removal of organic pollutants and nitrogen compounds with a two-stage treatment sequence (an activated sludge reactor in the first stage, and a trickling filter in the second stage), and with the application of floating carriers in the activated sludge reactor. The materials “Polyvom”, “Polystyrene” and “Bioballs” were adopted as floating carriers with previously determined filling ratios in the reactor volume of 10%, 20% and 20%, respectively. After the first stage of the study, it was found that the most effective treatment was achieved using the “Polyvom” material. Therefore, only this floating carrier was considered in the second and third stages of the study. Within the stages of the research, lab-scale benches operated under different operation modes of the treatment sequence. At the end of the study, it was possible to achieve the following levels of purification: BOD_5_ (2.1 mg/L), NH_4_ (0.4 mg/L), NO_2_ (1.0 mg/L), and NO_3_ (25 mg/L). The mean values of the concentrations of BOD, NH_4_, and NO_3_ met the requirements, but the concentration of NO_2_ exceeded the requirements (1.0 mg/L vs. 0.08 mg/L). These results were achieved under a hydraulic retention time in the activated sludge reactor of 8 h, and the MLSS for the free-floating and immobilized activated sludge was 0.2 and 0.9 g/L, respectively.

## 1. Introduction

The reduction in access to fresh water globally has already acquired serious proportions. This concerns even countries with a large available volume of fresh water sources, since if it is not so noticeable yet, consumption is growing, and the sources are not being replenished [1]. Therefore, the most serious issue on the agenda is the issue of water quality, which does not have a single and unambiguous solution. Obviously, optimization of water management is strongly required, which means optimization of water purification, distribution, and consumption, and optimization of wastewater treatment [2]. Balanced water management will certainly pave the way towards a sustainable urban water system considering technical, environmental, social, and economic aspects [3].

As mentioned above, optimization of wastewater treatment processes is required because wastewater after purification is discharged back into the aquifer, which, in turn, results in a significant influence of wastewater on the composition and quality of water in the aquifer. A serious problem for water bodies is eutrophication, which is caused by the presence of nitrogen (N) and phosphorus (P) compounds in the water. These two elements (both called nutrients), which appear in aquifers mostly with treated wastewater discharge, stimulate the growth of the flora of the reservoir, but these processes are destructive to the fauna [4,5].

Due to the peculiarities of water consumption (high water consumption rates, a high degree of infiltration into sewer collectors, frequent uncontrolled leaks through poor-quality water-collecting fittings), a significant proportion of treatment facilities (more than one third) receive low-concentrated wastewater, especially with regard to the content of organic pollutants [6]. As a result, applied treatment technologies should consider local conditions, which may make it impossible to apply many technological schemes due to the low ratio of organic pollutants in wastewater to the content of nitrogen compounds (BOD/N). Nitrogen compounds contained in household wastewater are divided into two categories: biodegradable and non-degradable [7]. Biologically non-degradable nitrogen is usually referred to as an inert fraction of COD that is not involved in biological purification processes. In most cases, biodegradable nitrogen from untreated wastewater is represented by ammonium nitrogen (N–NH_4_), as well as dissolved and suspended organic nitrogen fractions. The suspended fraction of organic nitrogen can be hydrolyzed into a dissolved fraction and then converted into the form of ammonium nitrogen by heterotrophic bacteria that use it as a biogenic source, and autotrophic nitrifying bacteria that use ammonium nitrogen as an energy source [8].

At most municipal wastewater treatment plants, a two-stage biological nitrogen removal process is traditionally used, including aerobic nitrification of ammonium nitrogen to nitrites (N–NO_2_) and nitrates (N–NO_3_) and subsequent denitrification of N-NO_x_ compounds to nitrogen gas (N_2_) [9].

Denitrification is carried out under anoxic conditions by heterotrophic denitrifying microorganisms, whose vital activity requires a source of biodegradable carbon (as an electron donor). Therefore, the efficiency and speed of denitrification in such schemes is strictly limited by the amount of available organic matter in the wastewater entering the treatment facilities [10].

Available experience reveals several possible directions for further research: the investigation of hydraulic retention time (HRT) [11,12], the influence of dissolved oxygen (DO) [13] or mixed liquor suspended solids (MLSS) on treatment processes; the enhancement of the treatment sequence; and the introduction of membrane modules to upgrade a conventional activated sludge reactor (ASR) into a membrane bioreactor (MBR). When domestic sewage is concerned, any enhancement of the treatment sequence aims to remove carbon (biochemical/chemical oxygen demand (BOD/COD)) and nutrients (nitrogen (N) and phosphorus (P)). 

To date, various treatment sequences have been developed using the principles of nitrification and denitrification [14,15], which are the key processes for nitrogen removal through biological methods. They can be systematized according to numerous approaches to their execution. 

Secondary treatment reactors can be designed as an ASR (Figure 1 and Figure 2) with trickling filters (TF, Figure 3), which can act both as nitrification and denitrification reactors [16,17]. According to the number of stages, treatment systems can be of a single-stage or two-stage [17,18,19]. Figure 1 shows examples of one- and two-stage nitrification systems in ASRs. In the case of industrial sewage, the number of stages can be more than two due to high concentrations of pollutants.

Figure 3 shows several schemes of treatment with TFs used for nitrification. These systems may also be considered as two-stage systems, although they may resemble single-stage systems since this type of reactor can be considered as a plug-flow reactor. However, the conditions necessary for the nitrification process are created only in the lower part of the TF [20,21,22]. The introduction of TFs may be quite reasonable as these facilities require less area and less energy for their operation, but TF can still provide relatively high quality treatment [23]. 

Although activated sludge treatment is undoubtedly more efficient than TF, trickling filters may be efficient enough to remove residual contamination with less demand for area and energy consumption in a multi-stage treatment [17,19]. In addition, trickling filters are initially installed at some wastewater treatment plants (WWTP). In this case, within the reconstruction, ASR is added to the technological scheme, and the trickling filter (usually after the replacement of loading material) is used as an additional purification facility, which allows for higher efficiency.

Another option for the operation of the WWTP can be a two-stage scheme using TF at the first stage and ASR at the second stage (Figure 4). In recent years, interest in TF+ASR facilities has increased, especially for the treatment of concentrated industrial effluents. Moreover, the application of lightweight polymer materials helps to replace traditional (and heavy) crushed stone carriers for the immobilization of biomass [24,25,26].

In various literary sources [27,28], it is noted that the introduction of N/D systems, as well as the intensification of these processes, can be widely used not only in the construction of new facilities but also in the reconstruction of existing plants. Designed more than a decade or even decades ago, both ASR and TF can no longer provide the treatment values required by the standards which have become stricter, especially concerning nutrients [29]. There are several reconstruction options in each case. At activated sludge facilities, depending on the specific conditions and requirements, various treatment sequences can be implemented to achieve the desired indicators [30]. In most TF, the loading material may be heavily clogged after years of operation, so the simplest reconstruction option is to replace it with a new material with better characteristics [31]. However, only the replacement of the loading material may not produce the desired effect unless a larger-scale, technological reconstruction is carried out with the introduction of new treatment technology. Another option for technological reconstruction is the inclusion of existing TF in multi-stage technological purification schemes that can meet the regulatory requirements for purification, both for BOD and nutrient removal. The combination of TFs and ASRs is considered promising [32].

The use of immobilized microflora, which may improve the elimination of both organic and nitrogen compounds from wastewater, is one potential strategies to intensify the process of advanced secondary treatment of wastewater [33]. The surface characteristics of suspended carriers are one of the important parameters impacting microbe adhesion to the carrier surface and subsequent biofilm formation in the MBBR/IFFAS process [34,35]. Most commercial carriers are now constructed of hydrophobic and negatively charged polymers such as polyethylene (PE) or polypropylene (PP). These polymers are often incompatible with the production of hydrophilic and electronegative biofilms [36].

Based on the above, this research was performed on three types of floating carriers (FC) applied in ASR, and the treatment sequence was enhanced with the addition of a TF. The research was carried out at Moscow State University of Civil Engineering on a lab-scale bench operating in flow-through mode.

## 2. Materials and Methods

The study presented in this paper has continued the previous research of the author on the application of floating carriers (FC) for biological wastewater treatment, which described the investigation of three types (Figure 5a–c) of FC made of polymer materials [37]. The previous results can be generally assessed as relatively positive, but the values of pollutant indicators exceeded the current limit values established for treated wastewater discharged into water bodies.

### 2.1. Research Principles

For this study, a research procedure was initially proposed in accordance with Figure 6a but within this approach, the results obtained could only be partially considered positive (see Section 3.1). Therefore, the research was continued, and the final frame of the research procedure is shown in Figure 6b. As can be seen, the study was divided into three stages, during which adjustments were made to the treatment process to achieve the best efficiency.

The principle of the research used earlier [1] was mostly continued:

1. Synthetic wastewater made on the basis of the following components was used for the study:-peptone enzymatic—110 g;-sodium chloride—68.8 g;-potassium nitric acid—1.2 g;-soda ash—16.2 g;

2. The synthetic wastewater solution was prepared in such a way as to ensure that its composition and the composition of real wastewater that is supplied for treatment to facilities in the Russian Federation are consistent.

3. Based on the above, the concentrations of BOD during the study were in the range of 100 to 250 mg/L, and the concentrations of ammonium nitrogen impurities ranged from 15 to 35 mg/L.

4. Sampling of wastewater throughout the study was carried out manually for the purpose of subsequent analysis based on standard methods used in Russia.

5. Wastewater samples were examined according to the following indicators: -Biological oxygen demand (BOD_5_);-Total suspended solids (TSS);-Ammonium nitrogen (NH_4_);-Nitrites (NO_2_);-Nitrates (NO_3_);-Dissolved oxygen (DO).

### 2.2. Bench Description

Within the research, a two-step sequence of treatment was implemented (Figure 7). The study considered a treatment sequence with an ASR (with a FC) in the first stage and a TF with a polymer loading material in the second stage.

The ASR model was made in the form of a cylinder with a height of 1 m, an internal diameter of 100 mm, and a volume of 8 L. The model of the TF had the same dimensions as the ASR (height of 1 m, inner diameter of 100 mm), with the filtering layer of the loading material of a height of 0.9 m.

The loading material consisted of plastic cylindrical elements with a polyethylene mesh coating (Figure 8). The laboratory TF model had a lower height of filtering layer compared with a conventional TF, which is normally 1.5–2 m. The TF model was almost isolated from air access, which created anoxic working conditions.

After treatment, the effluent was discharged to the secondary clarifier (SC), where the mixed liquid was divided by sedimentation into purified wastewater, which was discharged, and the biofilm, which was settled into the sedimentary cone and returned to ASR by means of an airlift. 

Synthetic wastewater was prepared in a plastic tank (volume 60 L) and supplied to the following treatment set-up. The ASR, TF, and SC were connected to each other with rubber hoses (diameter of 10 mm), which allowed the collection of samples. The ASR was supplied with air by a compressor. Air distribution was achieved by means of a ceramic aeration system, which provided fine-bubble aeration. The recirculation of the activated sludge was air-driven. 

## 3. Results and Discussion

### 3.1. Stage 1 of the Research

A key aim of the research was to maintain similar conditions on the three treatment benches. The wastewater flow rate was 0.04 m^3^ day^−1^; the flow rate of RAS was 0.1 m^3^ day^−1^ (recirculation rate—250%); and the HRT in the ASR was reduced from 8 to 5 h due to the addition of the second step of the treatment. The HRT in the SC was 2.5 h. 

In total, three benches were under simultaneous operation conditions, on which FC samples were examined in parallel. On bench 1, the ASR was equipped with the PS floating carrier with a filling ratio (FR) of 10%; the second bench was equipped with the PV floating material with an FR of 20%, and the BB floating carrier was installed on bench 3 with an FR of 20%. 

Figure 9, Figure 10, Figure 11 and Figure 12 show the results of chemical analyses of wastewater samples describing the operation of the FCs in combination with this technological scheme at stage 1.

As can be seen in Figure 9, the three FC achieved removal of organic pollution (BOD_5_) on a similar level of 80–90% with approximately the same average concentration of BOD_5_ in the influent. 

Despite the similar (and positive) results of the experiment regarding the removal of organic pollutants (Figure 9), the levels of purification achieved for ammonium nitrogen (Figure 10) for the three FC were significantly different. The highest efficiency was achieved with the PV floating carrier (almost 90%), and the lowest with the BB floating carrier with an average efficiency of 40%. Considering the absolute values, the NH_4_ mean concentrations in the effluent were 2.7 mg/L, 6.7 mg/L, and 14 mg/L for PV, PS, and BB, respectively. The overall ammonium oxidation was not sufficient to produce the required quality. 

In a previous study [1], attention was paid only to the removal of BOD_5_ and NH_4_. In this current research, NO_2_ and NO_3_ were also analyzed to estimate the completeness of nitrification and denitrification processes. However, Figure 11 and Figure 12 reveal that both processes were far from complete, especially concerning the removal of NO_2_ (Figure 11). Relatively high removal rates of NH_4_ (for the PV floating carrier) were not matched by any sustainable results for other forms of nitrogen.

Table 1 summarizes the average values for the removal of BOD and NH_4_, as well as the treatment efficiency in the activated sludge reactor. The combination of the two steps of treatment enabled a very high efficiency of BOD_5_ removal: 91%, 96%, and 95% for PS, PV, and BB, respectively. The two-step efficiencies for NH_4_ were 81%, 94%, and 61%, respectively. 

The best results at this stage, both from the viewpoint of treatment efficiency and in absolute values, were achieved in the unit equipped with PV. It is important to note that the results were obtained under a very low value of MLSS of free-floating sludge. At the same time, this FC provided the highest rate of attachment of microorganisms on its surface. In contrast, the BB floating carrier showed three times lower levels of attachment of sludge (0.3 g/L vs. 0.9 g/L), which probably led to poor treatment efficiency. 

However, as can be seen in Figure 10, even for the PV floating carrier the significant degree of ammonium nitrogen removal did not reach the values set by the standards (2.7 mg/L vs. /0.5 mg/L). There was also an excess nitrite concentration, which indicated the incompleteness of the nitrification process. Some instability in the ongoing treatment processes was also noted. In general, the interim results indicated positive dynamics but required improvement.

### 3.2. Stage 2 of the Research

In stage 2 of the research, the treatment sequence of the laboratory unit was changed. The flow of the RAS from the SC was redirected to the TF (instead of the ASR, which began operating without the return flow of activated sludge). The modified technological scheme is shown in Figure 13. The treatment scheme was altered for two reasons. First, the recirculating biomass coming from the SC could presumably create conditions for secondary contamination in the ASR, and, second, it was decided to maintain a relatively low value of MLSS in the ASR to confirm the results obtained in stage 1. An example of a technological scheme with an ASR, which operates without the return of activated sludge (Figure 2 and Figure 4) was described in [12,17]. It was also noted, that such a scheme can provide stable treatment operation with a minimal amount of excess activated sludge.

As indicated earlier, the lab bench equipped with a PV floating carrier showed generally better purification rates for BOD_5_ and NH_4_ in comparison to the other two units; therefore, further studies were carried out only with a PV floating carrier. The results of the analyses carried out at stage 2 are presented in Figure 14, Figure 15, Figure 16 and Figure 17.

Figure 14 and Figure 15, which show the results for BOD and NH_4_ removal, confirmed the assumption that the operation of ASR without a flow of RAS can make the process of treatment more stable, even taking into account that the average influent concentrations of BOD_5_ and NH_4_ in Stage 2 had increased significantly. The maximum values of BOD_5_ and NH_4_ in the influent were recorded at 240 mg/L and 31 mg/L, respectively (verses 164 and 21.3 mg/L, respectively, in stage 1). At the same time, the level of pollution removal in this stage increased, and the absolute values of the concentration in the effluent were lower than those in the previous stage. Taking into account the effect in the TF, the BOD_5_ and NH_4_ indicators met the requirements of the standards (2.1 and 0.5 mg/L, respectively). Thus, it can be concluded that the changes made to the operating mode of the installation had a positive impact on its efficiency.

Figure 16 shows that the concentration of nitrites after the TF still exceeded requirements of the standard, and remained at similar levels to those in stage 1; therefore, nitrification in the system still was not complete.

Figure 17 shows that content of nitrates in the effluent increased in stage 2, remaining within the limits (40 mg/L). Nevertheless, it is still urgent to focus on all three forms of nitrogen. The mean values of pollutant concentrations are summarized in Table 2.

### 3.3. Stage 3 of the Research

In order to achieve the complete removal of nitrogen (for NH_4_, NO_2_, and NO_3_), in stage 3, it was decided to return to the more common HRT value equal to 8 h in ASR. At the same time, the RAS flow was reduced from the previous value of 250% to 100%. The scheme of operation of the installation remained similar to that used in stage 2 (Figure 15). The results of the installation operation in stage 3 are shown in Figure 18, Figure 19, Figure 20 and Figure 21.

Figure 20 and Figure 21 illustrate the removal of organic pollutants and ammonium nitrogen, which are similar to the removal levels observed in stage 2. Therefore, the treatment process may be considered stable under various values of HRT. It should be pointed out that within the current research, higher efficiency was achieved with higher values of pollutants, which was also revealed by Chu and Wang in their research [38]. An obvious drop in the concentration of NO_2_ (see Figure 20) and a rise in the concentration of NO_3_ (see figures) in the stage 3 indicates better nitrification/denitrification conditions and intensity in the system, which resulted from both the redirected RAS flow and the higher value of HRT. This also speaks in favor of improved stability in the treatment process compared with the results in stages 1 and 2.

As can be seen from Table 3, in stage 3, the removal of BOD_5_ and NH_4_ remained at a consistently high level, and the absolute values of these indicators in the effluent met the required standards. Attention is drawn to the fact that the MLSS value for free-floating sludge in the system was significantly lower compared with stage 2, but this did not affect the quality of treatment. To summarize the efficiency in all three stages of the research, the removal of organic pollutants (BOD) and NH_4_ in the systems with floating carriers is well investigated under various conditions [39,40]. Redirection of the RAS flow in stages 2 and 3 had a positive impact on the efficiency of the treatment process (see Table 3), which rose from 90 to 95% (in the ASR), with an overall efficiency of 98–99% both for carbon and nitrogen removal.

However, despite earlier assumptions, the increase in HRT did not lead to a complete decrease in the concentration of nitrites to the standard values (Figure 20), whereas the concentration of nitrates met the standards (Figure 21). Similar results were observed in the research of Chen et al. [41]. This means that nitrification processes do not occur completely in this system, namely, nitrites do not completely turn into nitrates.

After the completion of the research, three samples of the FC were collected to evaluate their stability by analysis of the COD value. The COD indicates the destruction of the sample which may cause emission of unstable particles of the FC into wastewater. The sample of FC, which was not used for treatment, has a COD value of 2.4 mg/L [37]. After being used in the treatment processes, the samples were cleaned by washing out the remaining particles of the activated sludge. After cleaning, the FC samples were immersed for 2 h in clean water, which was then chemically analyzed. The results of this analysis show that the mean COD value for three samples was 2.9 mg/L. A slight increase in the COD may occur due to particles of the activated sludge remaining within the FC samples, which proves the stability of the floating carrier.

## 4. Conclusions

The following conclusions can be drawn based on the performed study: A comparative analysis of the operation in and ASR of three types of floating carriers showed that the highest efficiency was achieved when using the Polyvom material (91% and 89% for BOD and NH_4_, respectively). The other two FCs, Polystyrene and Bioballs, had comparable carbon removal efficiencies (81% and 88%) but significantly lower ammonium removal efficiencies (74% and 40%). Based on these results, the studies were continued. However, only the Polyvom FC was used in the later stages of the study.In stages 2 and 3 of the study, the treatment sequence was modified first by redirecting the RAS flow (to a TF instead of an ASR), and then by changing the hydraulic retention time in the ASR (from 5 to 8 h). The corrections enabled higher efficiency (98 and 99% for BOD and NH_4_) and improved the purification process stability (particularly nitrification and denitrification). According to the indicators of BOD, NH_4_, and NO_3_, the required values of pollution concentrations in discharged wastewater were achieved.It should be noted that the indicated treatment efficiency was obtained at low MLSS values (0.3 g/L for free-floating MLSS and 0.9 g/L for immobilized sludge). In the conditions of a real WWTP, such a mode of operation will reduce the costs (necessary facilities, energy consumption, reagents, etc.) for the treatment of excessive activated sludge.It is noted that despite the applied changes, it was not possible to achieve the required removal of nitrites from wastewater (1.0 vs. 0.08 mg/L), which is one of the promising areas for further research. In addition, it is of interest to study the work of floating carriers in membrane bioreactors. This combination may, on the one hand, increase the efficiency of cleaning processes; on the other hand, the floating carrier can be considered as a method to mitigate membrane fouling; the removal of the cake layer on the membrane surface occurs due to the friction of the floating carrier on the membrane surface [42,43].

## Figures and Tables

**Figure 1 polymers-14-02604-f001:**
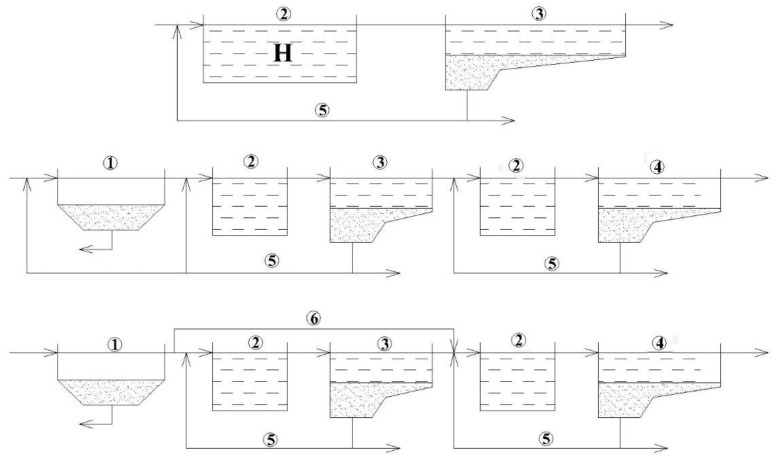
One-stage and two-stage activated sludge systems for nitrification: 1—primary clarifier (PC); 2—activated sludge reactor for nitrification (ASRN); 3—secondary clarifier (SC); 4—tertiary clarifier (TC); 5—return activated sludge flow (RAS); 6—influent split for secondary treatment.

**Figure 2 polymers-14-02604-f002:**
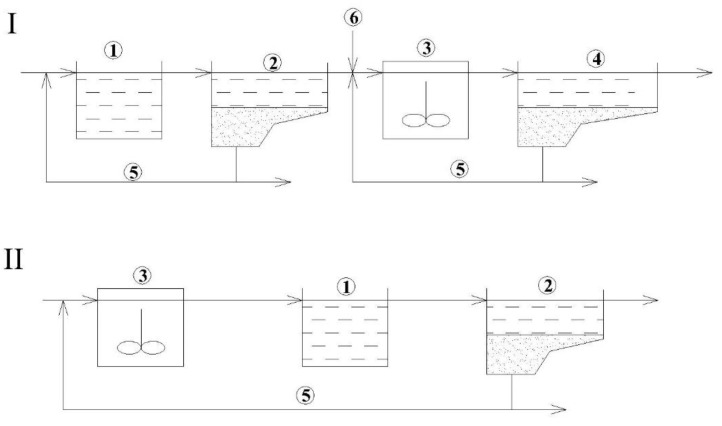
Two-stage nitrification/denitrification (N/D) systems: 1—ASRN; 2—SC; 3—Active sludge reactor for denitrification (ASRD); 4—tertiary clarifier (TC); 5—RAS; 6—carbon feed (CF).

**Figure 3 polymers-14-02604-f003:**
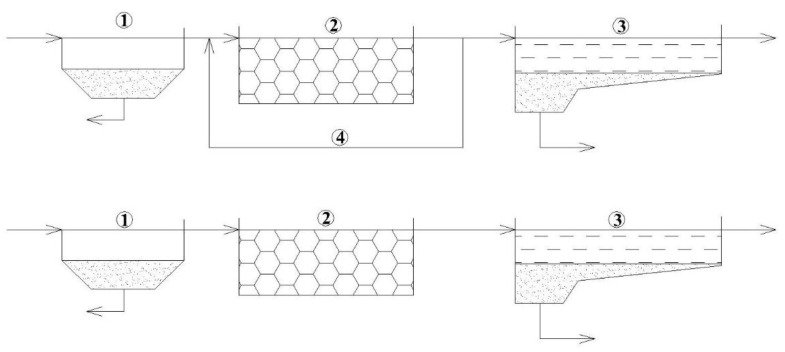
Trickling filters schemes: 1—PC; 2—TF; 3—SC; 4—recycle flow (RF).

**Figure 4 polymers-14-02604-f004:**
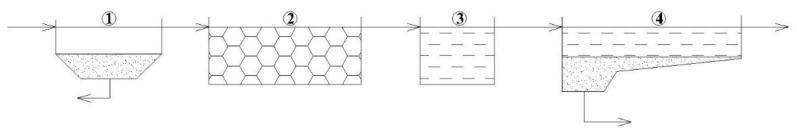
Two-stage nitrification system: 1—PC; 2—TF; 3—ASRN; 4—SC.

**Figure 5 polymers-14-02604-f005:**
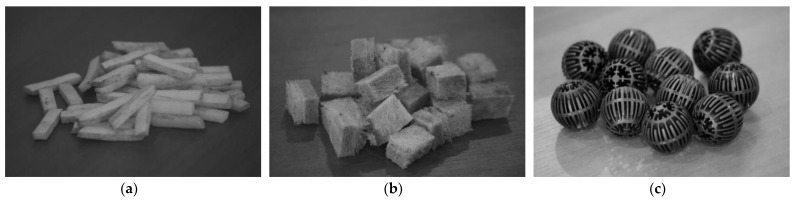
Floating carriers used in the research: (**a**) Polystyrene; (**b**) Polyvom; (**c**) Bioballs.

**Figure 6 polymers-14-02604-f006:**
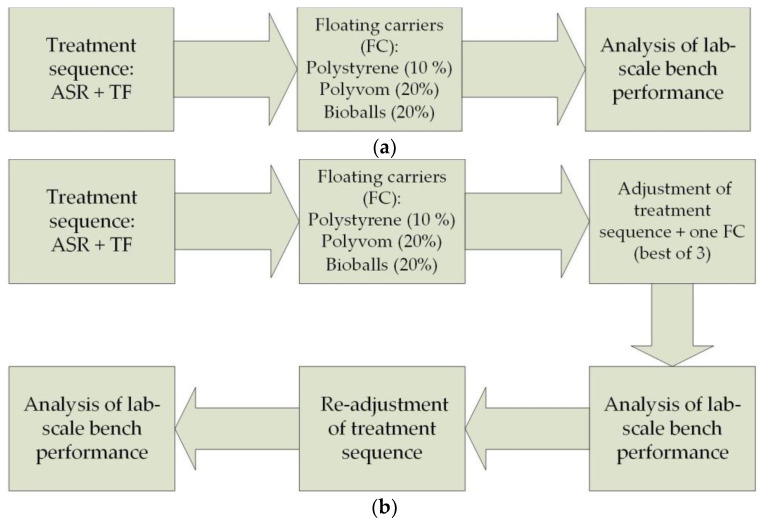
Research sequence: (**a**) initial research procedure: (**b**) final research procedure.

**Figure 7 polymers-14-02604-f007:**
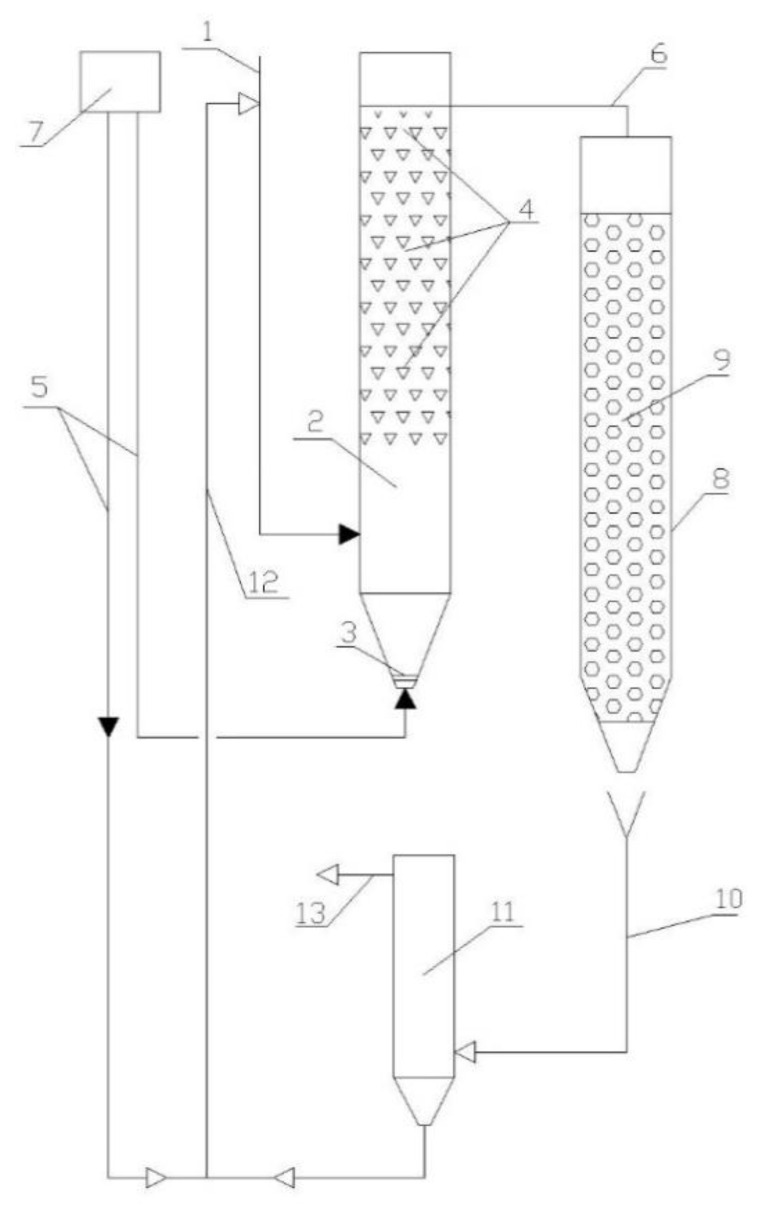
Scheme of the lab-scale bench: 1—wastewater influent; 2—ASR; 3—aeration device; 4—floating carrier; 5—air supply; 6—effluent from ASR to TF; 7—compressor; 8—TF; 9—biomass carrier in TF; 10—discharge of treated wastewater for clarification; 11—SC; 12—RAS; 13—effluent.

**Figure 8 polymers-14-02604-f008:**
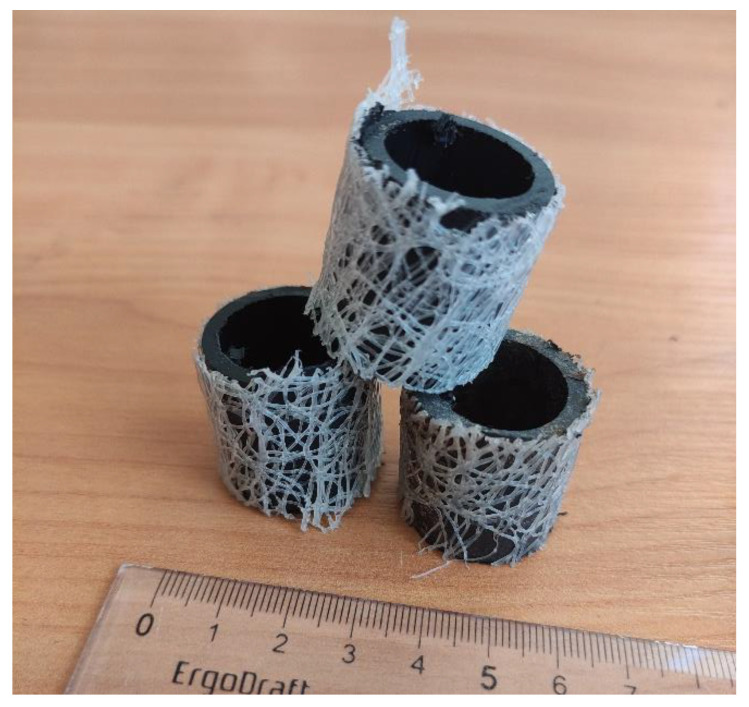
TF Filter elements.

**Figure 9 polymers-14-02604-f009:**
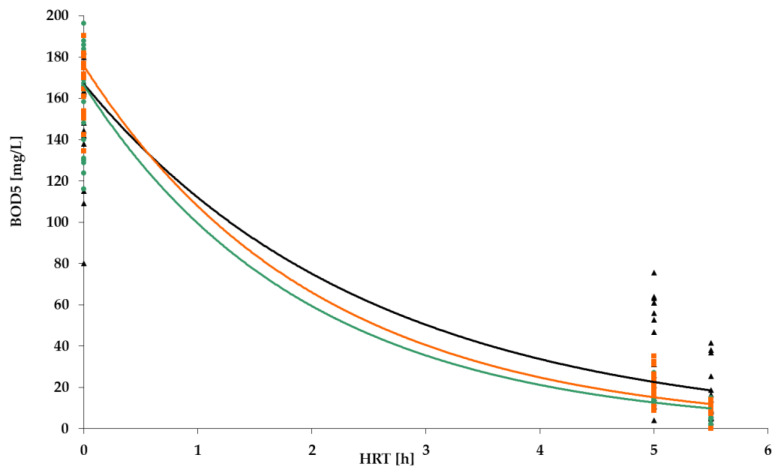
BOD removal dynamics for: PS floating carrier (black); PV floating carrier (green); BB floating carrier (orange).

**Figure 10 polymers-14-02604-f010:**
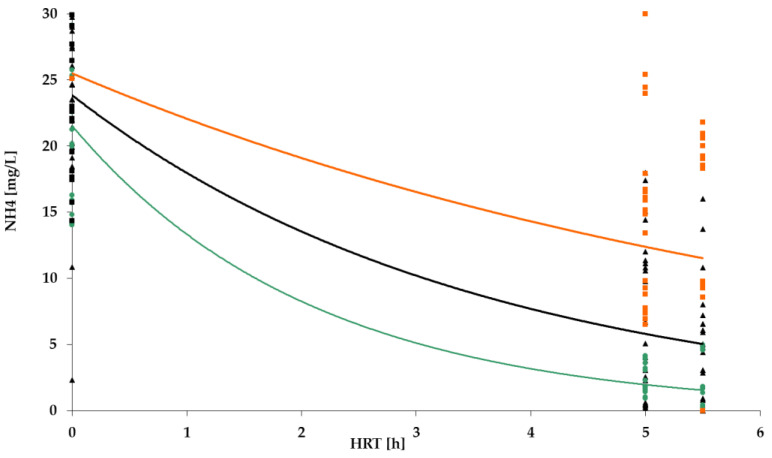
NH_4_ removal dynamics for: PS floating carrier (black); PV floating carrier (green); BB floating carrier (orange).

**Figure 11 polymers-14-02604-f011:**
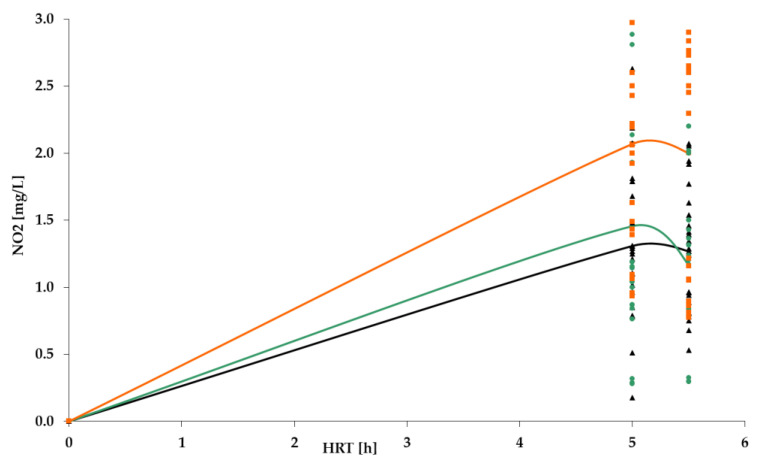
NO_2_ concentration dynamics for: PS floating carrier (black); PV floating carrier (green); BB floating carrier (orange).

**Figure 12 polymers-14-02604-f012:**
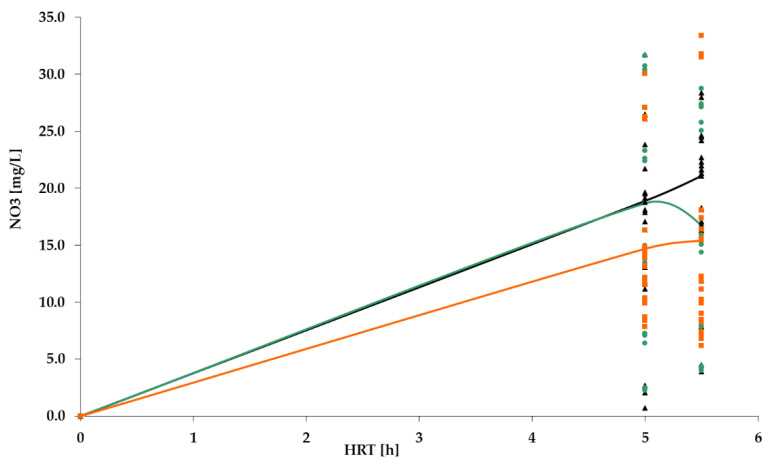
NO_3_ concentration dynamics for: PS floating carrier (black); PV floating carrier (green); BB floating carrier (orange).

**Figure 13 polymers-14-02604-f013:**
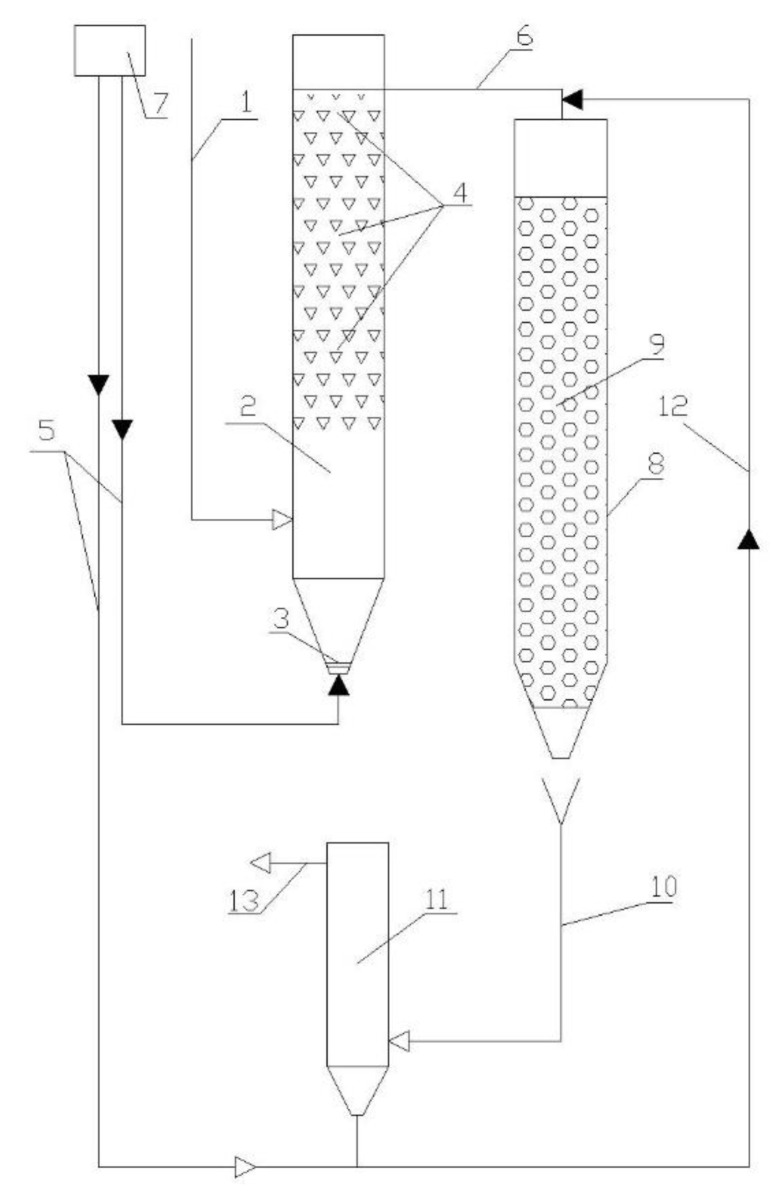
Scheme of the lab-scale bench in stage 2: 1—wastewater influent; 2—ASR; 3—aeration device; 4—floating carrier; 5—air supply; 6—effluent from ASR to TF; 7—compressor; 8—TF; 9—biomass carrier in TF; 10—discharge of treated wastewater for clarification; 11—sedimentation tank; 12—RAS; 13—effluent.

**Figure 14 polymers-14-02604-f014:**
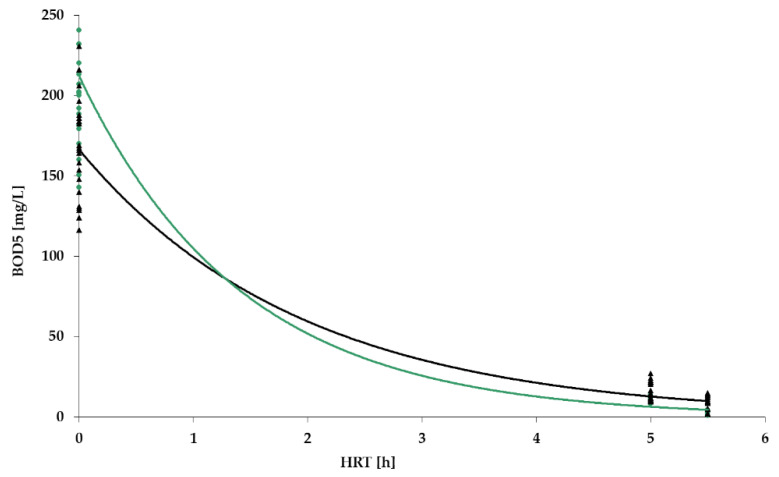
BOD_5_ removal dynamics for PV floating carrier: stage 1 (black); stage 2 (green).

**Figure 15 polymers-14-02604-f015:**
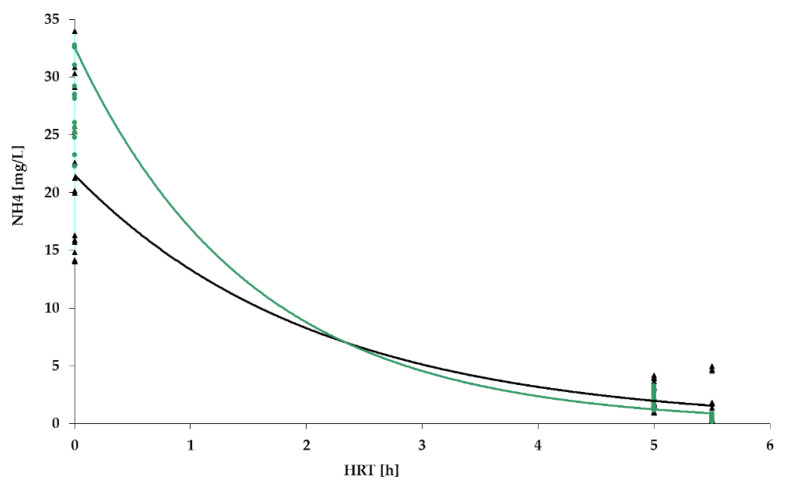
NH_4_ removal dynamics for PV floating carrier: stage 1 (black); stage 2 (green).

**Figure 16 polymers-14-02604-f016:**
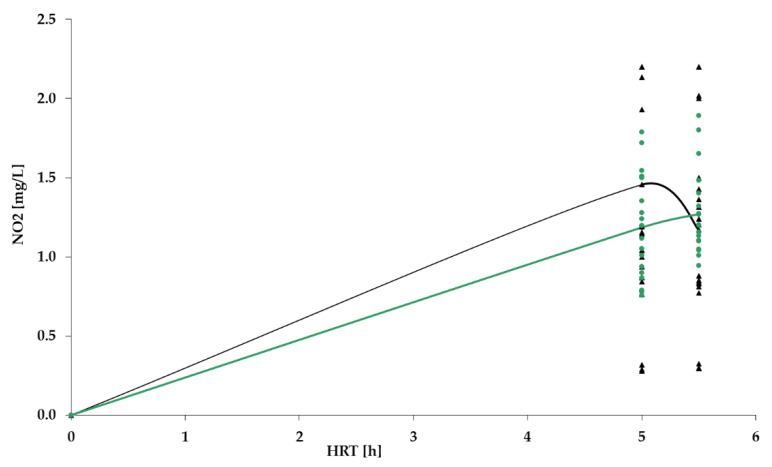
NO_2_ dynamics for PV floating carrier: stage 1 (black); stage 2 (green).

**Figure 17 polymers-14-02604-f017:**
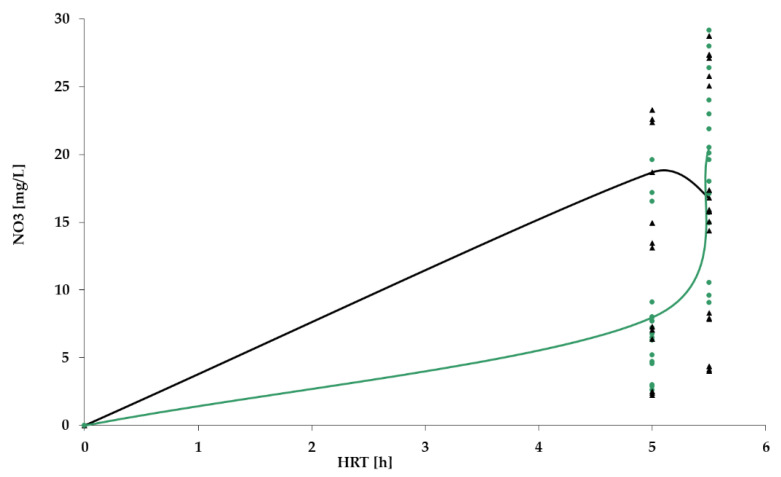
NO_3_ dynamics for PV floating carrier: stage 1 (black); stage 2 (green).

**Figure 18 polymers-14-02604-f018:**
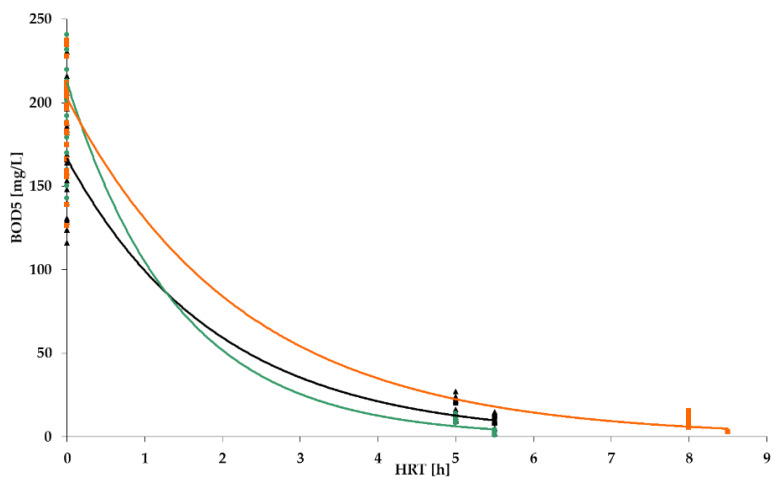
BOD_5_ removal dynamics: stage 1 (black); stage 2 (green); stage 3 (orange).

**Figure 19 polymers-14-02604-f019:**
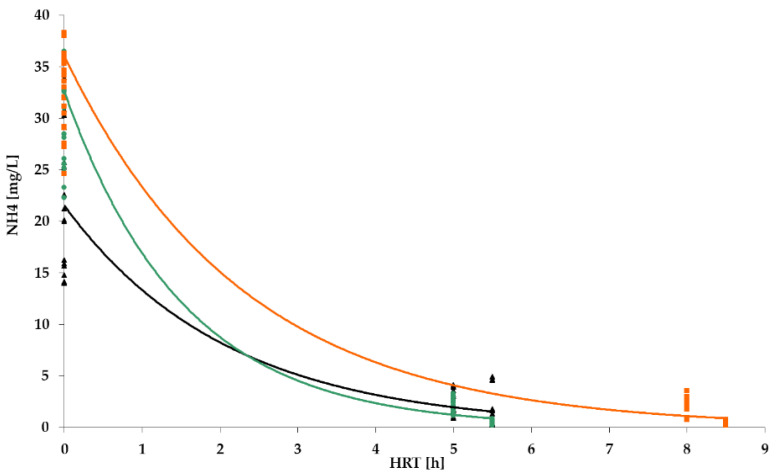
NH_4_ removal dynamics: stage 1 (black); stage 2 (green); stage 3 (orange).

**Figure 20 polymers-14-02604-f020:**
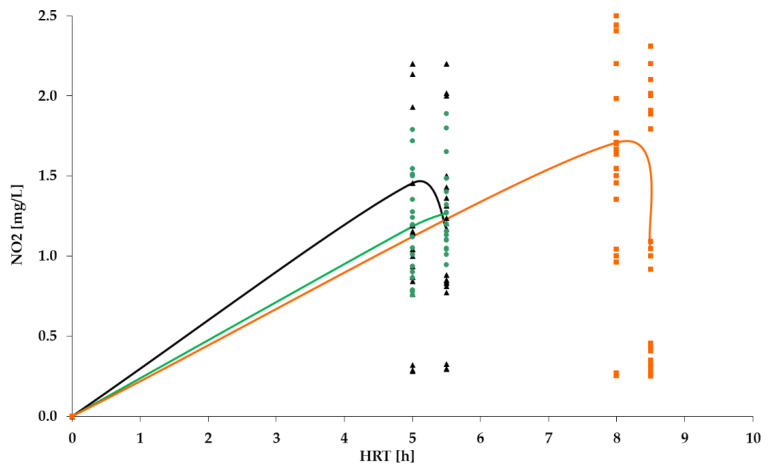
NO_2_: stage 1 (black); stage 2 (green); stage 3 (orange).

**Figure 21 polymers-14-02604-f021:**
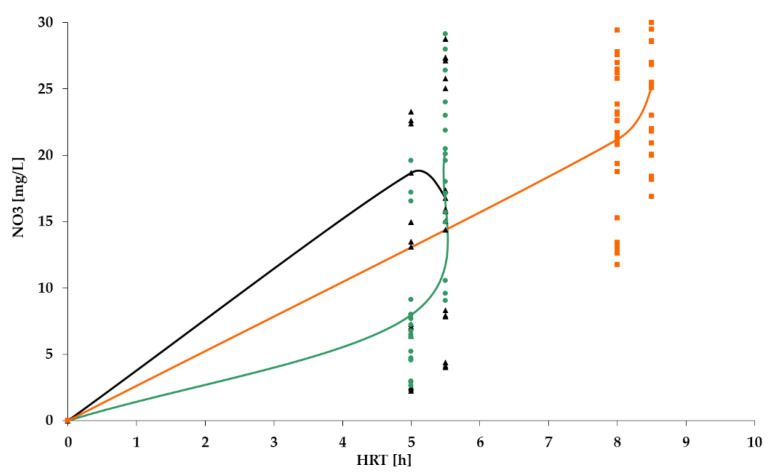
NO_3_ dynamics: stage 1 (black); stage 2 (green); stage 3 (orange).

**Table 1 polymers-14-02604-t001:** Treatment indicators at stage 1.

Type of FC	Mean Values [mg/L]						MLSS	
	BOD_5_			NH_4_			[g/L]	
	In	Out	Eff %	In	Out	Eff %	Free	Att.
PS	162	31	81	23.5	6.7	74	0.84	0.2
PV	164	14.6	91	21.3	2.7	89	0.15	0.9
BB	171	21.1	88	25	14	40	0.4	0.3

**Table 2 polymers-14-02604-t002:** Comparison of treatment process at stage 1 and 2 of the research.

Mean Values		Stage 1		Stage 2	
BOD_5_	Influent [mg/L]	164		202	
	After ASR [mg/L]	14		10	
	After TF [mg/L]	6.3		2.1	
	Efficiency [%] *	91	96	95	98
NH_4_	Influent [mg/L]	21.3		31	
	After ASR [mg/L]	2.3		2.1	
	After TF [mg/L]	1.4		0.5	
	Efficiency [%]	89	94	93	98
NO_2_	After ASR [mg/L]	1.5		1.2	
	After TF [mg/L]	1.2		1.3	
NO_3_	After ASR [mg/L]	18.7		8.0	
	After TF [mg/L]	16.8		20.5	
MLSS [g/L]	Free floating	0.15		0.3	
	Attached	0.9		0.9	

* Note—Here and in following sections, the treatment efficiency is given both for ASR only, and for the combination ASR + TF.

**Table 3 polymers-14-02604-t003:** Comparison of treatment processes in stages 1, 2, and 3 of the research.

Mean Values		Stage 1		Stage 2		Stage 3	
BOD_5_	Influent [mg/L]	164		202		197	
	After ASR [mg/L]	14		10.6		10.1	
	After TF [mg/L]	6.3		2.1		2.0	
	Efficiency [%]	91	96	95	99	95	99
NH_4_	Influent [mg/L]	21.3		31		34	
	After ASR [mg/L]	2.3		2.1		2.2	
	After TF [mg/L]	1.4		0.5		0.5	
	Efficiency [%]	89	94	93	98	94	99
NO_2_	After ASR [mg/L]	1.5		1.2		1.7	
	After TF [mg/L]	1.2		1.3		1.0	
NO_3_	After ASR [mg/L]	18.7		8.0		21.2	
	After TF [mg/L]	16.8		20.5		25.1	
MLSS [g/L]	Free floating	0.15		0.3		0.1	
	Attached	0.9		0.9		0.9	

## Data Availability

Not applicable.

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
