# Peer review of "Advanced Research on Polymer Floating Carrier Application in Activated Sludge Reactors"

_polymers, 2022, doi:10.3390/polym14132604_

Round 1
Reviewer 1 Report
-The structure of this article is unclear.
-The writing style is not like writing other scientific articles.
- The introduction is not written correctly. In the introduction, it is better to provide explanations about past studies and the importance of the subject under study.
- The SR process seems to have been more successful than other processes, because the values ​​of most of the parameters measured at this stage have been significantly reduced compared to the initial stage and after TF, which needs to be explained in this section. What are the reasons for the results?
- In Table 3, the Efficiency section needs further explanation. Explain the table below.
- In Figures 22 and 23, sudden peaks occurred during the retention times of more than 5, but no specific explanation has been given.
- Conclusions to be corrected and completed.
- In different parts of the text of the article, it is explained about the working method, which is not correct, and the working method should be presented only in the materials and methods section.
Author Response
I would like to thank you for your comments, which helped to improve the quality of the article. Below is a detailed description of the comments and their consideration in the article.
Comment 1: The structure of this article is unclear.
Reply on the comment 1: The article has been re-arranged
Comment 2: The writing style is not like writing other scientific articles.
Reply on the comment 2: The language and styling of the manuscript has been revised
Comment 3: The introduction is not written correctly. In the introduction, it is better to provide explanations about past studies and the importance of the subject under study.
Reply on the comment 3: The Introduction section has been extended and re-arranged.
Comment 4: The SR process seems to have been more successful than other processes, because the values ​​of most of the parameters measured at this stage have been significantly reduced compared to the initial stage and after TF, which needs to be explained in this section. What are the reasons for the results?
Reply on the comment 4: The following text was added to the paper:
Although activated sludge treatment is undoubtedly more efficient than TF, trick-ling filters may be efficient enough to remove residual contamination with less demand for occupied area and energy consumption in multi-stage treatment [17, 19]. In addition, trickling filters were initially installed at some WWTP. In this case, within the reconstruction, ASR is added to the technological scheme, and the trickling filter (usually after replacement of loading material) is used as an additional purification facility, which allows for higher efficiency.
Comment 5: In Table 3, the Efficiency section needs further explanation. Explain the table below.
Reply on the comment 5: The following text was added to the paper:
Redirection of the RAS flow on the stages 2 and 3 had a positive impact on the efficiency of the treatment process (see table 3), which has risen from 90 to 95% (in the ASR) with overall efficiency of 98-99% both for carbon and nitrogen removal.
Comment 6: In Figures 22 and 23, sudden peaks occurred during the retention times of more than 5 but no specific explanation has been given.
Reply on the comment 6: The following text was added to the paper:
An obvious drop for concentration of NO2 (see figure 20) and rise of concentration of NO3 (see figures) at the stage 3 witnesses better nitrification/denitrification condition and intensity in the system, which happened due to both redirected RAS flow and higher value of HRT. This also speaks in favor of better stability of the treatment process.
Comment 7: Conclusions to be corrected and completed.
Reply on the comment 7: Conclusions section has been completely rewritten
Comment 8: In different parts of the text of the article, it is explained about the working method, which is not correct, and the working method should be presented only in the materials and methods section.
Reply on the comment 8: The article was re-arranged according to the recommendations of the reviewer. So now the explanation of the research left only in the Materials and methods section
The paper was significantly extended in order to follow the recommendations of reviewers.
Thank you for your consideration of this manuscript.
Reviewer 2 Report
This study reported the design of polymer floating carrier (three polymer biomass carriers) used in the activated sludge reactor for urban wastewater treatment through removing of carbon and nutrients.
There are many comments should be addressed as follows:
1- Regarding the abstract section:
This part is long and the author has provided many details and clarifications that can be highlighted in the introduction part.
Thus, this part should be carefully reviewed, highlighting the novelty of the work, the main objective and the main results obtained.
2- Regarding the introduction par:
a- First of all, this part needs careful revision as follows:
i- This section is short and lacks information and background related to the current study that will benefit the readers
ii- Moreover, there are many information about the author's previous work that prompt the reader to take the basic information from this published work.
iii- In this regard, the introduction part looks like a report on previous work.
b- Figures 1-3 are shown in previous works, why are they important and where are the copyrights?
** In case of these figures need to be displayed, they should be combined into one figure as a, b, c and not in three figures
c- More information and details associated with the main problem with brief information, the existing challenges, the available solutions, and the importance of the work should be highlighted
d- More attention should be paid to the design of the materials used and the efforts made in the field related to the current study. In addition to highlighting the advantages of the current study.
e- The author should clearly highlight the novelty of the materials used and the reason for choosing this composition
f- More details should be added to the claim presented at the end of this introduction with explaining the method of preparation, highlighting the novelty of the work, the novelty of the approach and technology used, the objective, innovation and key findings.
3- In the results and discussion
a- In general, the results and discussion part related to system design, novelty, and analysis is short and needs further explanation and discussion.
b- Furthermore, the author should provide further explanation and discussion about the results and data obtained to confirm the effectiveness of his system used as well as help the readers
c- Is there any analysis confirm the materials stability and structure
d- The conclusion part should be carefully revised with focusing on highlighting the technology used, work novelty, main objective, as well as clarification of the future and applicability. Also, it is preferable to submit the conclusion part as a single paragraph
e- The figure captions is not enough to describe the figure, the author must provide more information and details enough to clarify the figures.
f- The author should provide all figures with high resolution.
Author Response
Dear reviewer,
I would like to thank you for your comments, which helped to improve the quality of the article. Below is a detailed description of the comments and their consideration in the article.
1- Regarding the abstract section:
Comment: This part is long and the author has provided many details and clarifications that can be highlighted in the introduction part.
Thus, this part should be carefully reviewed, highlighting the novelty of the work, the main objective and the main results obtained.
Reply on the comment: The abstract has been rewritten according to the reviewers’ comments.
2- Regarding the introduction par:
Comment “a”: First of all, this part needs careful revision as follows:
i- This section is short and lacks information and background related to the current study that will benefit the readers
ii- Moreover, there are many information about the author's previous work that prompt the reader to take the basic information from this published work.
iii- In this regard, the introduction part looks like a report on previous work.
Reply on comment “a”: The Introduction section has been extended and re-arranged according to the comments of the reviewers
Comment “b”: Figures 1-3 are shown in previous works, why are they important and where are the copyrights?
** In case of these figures need to be displayed, they should be combined into one figure as a, b, c and not in three figures
Reply on comment “b”: The author captured figures 1-3 (photos) by himself, so no copyrights required. Figures 1-3 were combined into figure 5a-c and moved to “materials and methods with appropriate change of numbering of the previous and the following figures. I suppose it is necessary to present these photos also here to make the description of the research complete.
Comment “c”: More information and details associated with the main problem with brief information, the existing challenges, the available solutions, and the importance of the work should be highlighted
Reply on comment “c”: The description of the main problem of the research was extended in the introduction part
Comment “d”: More attention should be paid to the design of the materials used and the efforts made in the field related to the current study. In addition to highlighting the advantages of the current study.
Reply on comment “d”: The description of each floating carrier has been given in the previous paper of the author. It was decided not to overload the current paper, so the design of the materials was described in brief.
Comment “e”: The author should clearly highlight the novelty of the materials used and the reason for choosing this composition
Reply on comment “e”: As it was said in the previous comment, the reason for choosing of these materials were presented in the previous paper
Comment “f”: More details should be added to the claim presented at the end of this introduction with explaining the method of preparation, highlighting the novelty of the work, the novelty of the approach and technology used, the objective, innovation and key findings.
Reply on comment “f”:
3- In the results and discussion
Comment “a”: In general, the results and discussion part related to system design, novelty, and analysis is short and needs further explanation and discussion.
Reply on comment “a”: the analysis throughout the manuscript of the results has been extended
Comment “b”: Furthermore, the author should provide further explanation and discussion about the results and data obtained to confirm the effectiveness of his system used as well as help the readers
Reply on comment “b”: The results explanation has been extended according to the recommendations of the reviewers.
Comment “c”: Is there any analysis confirm the materials stability and structure
Reply on comment “c”: The analysis was added in the end of results section (see text below)
After the completion of the research, three samples of the FC have been collected to evaluate their stability by analysis of COD value. COD may show the destruction of the sample which may cause emission of unstable particles of the FC into wastewater. The sample of FC, which was not used for treatment, has a COD value of 2.4 mg/L [37]. The samples after being applied for treatment were cleaned by washing out remaining particles of the activated sludge. After cleaning, FC samples were immersed for 2 hours into clean water, which was in the following chemically analyzed. The results of the analysis showed that the mean COD value for three samples was 2.9 mg/L. Slight grow of the COD may appear due to remaining particles of the activated sludge within the FC samples that proves the stability of the floating carrier.
Comment “d”: The conclusion part should be carefully revised with focusing on highlighting the technology used, work novelty, main objective, as well as clarification of the future and applicability. Also, it is preferable to submit the conclusion part as a single paragraph
Reply on comment “d”: The conclusion was initially presented as a single paragraph. According to the remarks of the reviewers, the conclusions section has been completely rewritten
Comment “e”: The figure captions is not enough to describe the figure, the author must provide more information and details enough to clarify the figures.
Reply on comment “e”: Figure captions were revised end extended
Comment “f”: The author should provide all figures with high resolution.
Reply on comment “f”: The figures requested by the reviewer are provided
The paper was significantly extended in order to follow the recommendations of reviewers
Thank you for your consideration of this manuscript.
Round 2
Reviewer 1 Report
The changes made have improved the quality of the article. Thankful
Reviewer 2 Report
The revised manuscript shows that most of the comments raised have been covered, therefore, my recommendation to accept this work in its current form